# Honey and Alzheimer’s Disease—Current Understanding and Future Prospects

**DOI:** 10.3390/antiox12020427

**Published:** 2023-02-09

**Authors:** Ammara Shaikh, Fairus Ahmad, Seong Lin Teoh, Jaya Kumar, Mohamad Fairuz Yahaya

**Affiliations:** 1Department of Anatomy, Faculty of Medicine, Universiti Kebangsaan Malaysia, Kuala Lumpur 56000, Malaysia; 2Department of Physiology, Faculty of Medicine, Universiti Kebangsaan Malaysia, Kuala Lumpur 56000, Malaysia

**Keywords:** honey, natural products, Alzheimer’s disease, oxidative stress, anti-oxidants, neuroprotection, flavonoids, phenolic acids

## Abstract

Alzheimer’s disease (AD), a leading cause of dementia, has been a global concern. AD is associated with the involvement of the central nervous system that causes the characteristic impaired memory, cognitive deficits, and behavioral abnormalities. These abnormalities caused by AD is known to be attributed by extracellular aggregates of amyloid beta plaques and intracellular neurofibrillary tangles. Additionally, genetic factors such as abnormality in the expression of APOE, APP, BACE1, PSEN-1, and PSEN-2 play a role in the disease. As the current treatment aims to treat the symptoms and to slow the disease progression, there has been a continuous search for new nutraceutical agent or medicine to help prevent and cure AD pathology. In this quest, honey has emerged as a powerful nootropic agent. Numerous studies have demonstrated that the high flavonoids and phenolic acids content in honey exerts its antioxidant, anti-inflammatory, and neuroprotective properties. This review summarizes the effect of main flavonoid compounds found in honey on the physiological functioning of the central nervous system, and the effect of honey intake on memory and cognition in various animal model. This review provides a new insight on the potential of honey to prevent AD pathology, as well as to ameliorate the damage in the developed AD.

## 1. Introduction

Alzheimer’s disease (AD) is a neurodegenerative disorder associated with damage to the brain areas such as the cerebral cortex, temporal lobe, hippocampus, amygdala, entorhinal cortex (EC), and parahippocampal region [1,2]. The disease is mainly characterized by impaired memory and cognitive deficits [3,4]. AD is considered the most common cause of dementia, accounting for about 60–70% of the total cases worldwide [5]. In addition to the deficits of memory and cognition, AD is also accompanied by behavioral changes. Since most of the areas affected by the pathology are involved both in cognition and behavior, the predominant behavioral changes, such as agitation, dysphoria, and apathy, are correlated highly with cognitive dysfunction [6].

Previous studies have proposed several effective solutions to reduce the deposition of amyloid fibrils, minimize oxidative stress and neuroinflammation, and/or improve memory and cognition. These can be divided into drugs and antioxidants or neuroprotective agents for ease of discussion. The drugs include N-methyl-d-aspartate (NMDA) receptors antagonists [7,8], agents acting on the acetylcholinergic system (ACh system) [9,10], anti-amyloid [11], and anti-tau [12]. The antioxidants or neuroprotective agents include idebenone (an organic compound from the quinone family) and α-tocopherol [13], estrogen analogues [14,15], and honey [16,17,18]. Although the allopathic medications have shown promising results in attenuating symptoms, they may cause a number of adverse effects while having some serious precautions and contraindications [10,19,20,21,22,23,24,25]. Comparably, honey only has one serious adverse effect that ranges from mild hypersensitivity reaction to anaphylactic shock, which is attributed to the presence of pollens and bee-derived proteins in honey [26]. These allergic reactions, however, are very rare with only a small number of cases reported till date [26,27,28]. Additionally, among all the above mentioned substances, honey stands out by having the potential to improve almost all aspects of AD, such as oxidative stress [29,30,31], neuroinflammation [32,33], neuroprotection [34,35], ACh system [36], and memory and cognition [37,38].

As AD is associated with the involvement of the central nervous system (CNS) that causes the characteristic impaired memory, cognitive deficits, and behavioral abnormalities, in this review article, we will limit our discussion of the effects of honey on the said aspects.

## 2. Pathophysiology and Clinical Picture of Alzheimer’s Disease

AD is believed to begin and caused by the accumulation of amyloid beta (Aβ) plaques; this perspective of AD progression is known as the amyloid-cascade-hypothesis [39]. According to this hypothesis, the neuropathology in AD starts from the extracellular accumulation of Aβ fibrils as abnormal neuritic plaques, the deposition of which leads to oxidative damage and inflammation. Although this is a widely accepted hypothesis, some researchers believe in the tau hypothesis, according to which tau pathology is a prerequisite for the Aβ aggregation to take place [40,41,42]. Additionally, there is a third viewpoint which suggests that there may be more than one pathological pathway co-occurring, as dementia in AD is not correlated with either plaque or tangle burden but with the serum amyloid protein content in the Aβ plaques [43,44]. This hypothesis is further supported by the findings that the neurofibrillary tangles (NFT)-bearing neocortical neurons are functionally intact [45], and even though the cognitive deficits increase with ageing, the load of neuritic plaques and NFTs tend to decline as the elderly people age, i.e., more burden in the 60–80-year-old individuals than in over 90-years old individuals [46].

Pathologically, AD is characterized by the deposition of Aβ plaques (extracellularly) and NFTs (intracellularly). Soon after the tau fibrils are hyperphosphorylated, they may be converted into pathological tau and result in the formation of NFTs [47]; the latter more commonly affects the medial limbic structures (MLS) comprised of hippocampus, subiculum, EC, and amygdala [48,49]. The tau aggregates need the presence of neuritic plaques, therefore, are likely formed adjacent to them [47], whereas NFTs have an independent presence [49]. Irrespective of the site of impaction of tangles, which may vary in the brain, the formation of the NFTs is possibly the result of the interplay of oxidative injury, neuroinflammation, ineffective degradation, and subsequent ubiquitination causing hyperphosphorylation of tau followed by the subsequent formation of tangles [50,51,52,53].

Considering the defect at the genetic level, AD results from the abnormality in the expression of five genes: Apolipoprotein E (APOE), Amyloid Precursor Protein (APP), Beta-site Amyloid precursor protein Cleaving Enzyme 1 (BACE1), Presenilin 1 and 2 (PSEN-1, and PSEN-2). While the pathology of the first gene APOE (especially the allelic variant ε4) is associated with sporadic AD [54], the latter four genes were found to be responsible for the familial AD. The most common form of AD is sporadic and its risk increases with the presence of ε4 allele [55,56]. The other alleles, i.e., ε2 and ε3, minimize oxidative damage and neuronal death, whereas, the ε4 allele has the lowest capacity to prevent cellular toxicity [57]. Therefore, its presence increases the likelihood of developing AD [57,58]. As for the familial AD, the APP gene is located on chromosome 21 and is responsible for the production of APP, which is required for the normal regulation of several cellular functions [59]; however, excessive dose, hence over-expression, of this gene results in increased amyloid levels in brain and likelihood to develop AD [60,61], as also observed in Down syndrome (trisomy 21) [62]. The other genes, BACE1 and PSEN (1 and 2), also known as β-secretase and γ-secretase, further play their part in AD pathogenesis [63,64,65]. It occurs when the APP is cleaved by BACE1 (β- secretase) instead of the normal cleavage by α-secretase, and the product acts as a substrate for γ-secretase resulting in the formation, and subsequently, aggregation of the Aβ oligomers [66,67]. Furthermore, since the β-secretase and γ-secretase act on the common substrate, the elevated level and activity of the former is mostly accompanied by a reduction in the level of the latter [68].

Overall, the damage in the brain in AD is comprised of injuries on both macroscopic and microscopic levels. The gross changes consist of a reduction of the total brain tissue with an increase in the volume of the ventricles [69,70], whereas the underlying microscopic changes include the loss of synapses [71,72], damage to pyramidal neurons, and neurodegeneration [1,73]. In addition, the loss of synapses can either occur in the presence of normal long-term potentiation (LTP) [74] or is probably due to impaired LTP [75,76]. To further shed the light on these contrasting results, recent studies described that AD may affect LTP in some pathways (e.g., Schaffer collateral) while the LTP in other pathways (e.g., mossy fibers) remain unaffected/normal [77] with a possible alteration in the short-term potentiation [78]. Moreover, AD brains are affected by oxidative injury and inflammatory damage. The former is due to an imbalance between antioxidants and oxidation-causing substances (i.e., free radicals and reactive oxygen species), causing a reduction in the activity of antioxidants such as superoxide dismutase (SOD), glutathione (GSH), and catalase, together with an increase in the markers of oxidative damage such as Malondialdehyde (MDA) (the product of lipid peroxidation) and 3-nitrotyrosine (the end product of protein oxidation), and 8-hydroxydeoxyguanosine and 8-hydroxyguanosine (the product of oxidation of guanine in DNA) [79,80,81]. Likewise, perpetual neuroinflammation marked by an imbalance in the inflammatory cytokines is also evident by the over-expression of the pro-inflammatory markers, such as IL-1α, IL-1β, IL-6 [82,83], TNFα, and NFκB [84,85], and an accompanied under-expression of some anti-inflammatory cytokines, such as IL-4 and IL-10 [86,87]. Moreover, the reduced level of anti-inflammatory cytokines further leads to the uninhibited activity of pro-inflammatory cytokines [87,88,89] and results in more neuronal damage [87]. Furthermore, an altered interaction of cytokines (both pro-inflammatory and anti-inflammatory) has been observed based on the underlying pathology of APOE genotypes [90,91].

Those mentioned structural and functional abnormalities that are the characteristic features of AD can start appearing in the brain in middle-aged individuals (familial or early-onset AD) or the elderly (sporadic or late-onset AD). Irrespective of the age of onset, clinically, AD presents as deficits of memory, cognition [48,92], and behavior [6].

## 3. Honey and Its Powerful Ingredients—The Phenolic Compounds

Honey mainly contains sugar and water [93]. The high sugar content, comprised of dextrose, levulose, and other complex carbohydrates, makes it a better alternative to glucose as it replenishes energy with a constant blood glucose level [94]. In addition to being a mixture of around 30 different kinds of sugars [95], honey has several minor components, including phenolic compounds, proteins, amino acids, vitamins, enzymes, and minerals [93,96]. Although the main constituents (water and sugars) remain the same, the composition of minor components of each honey type varies significantly, which is due to the difference in geographical location, floral source, storage, and the final color [95,97]. Due to the mentioned factors, various types of honey are different in composition of polyphenolic compounds [98,99], and therefore, polyphenolic activity and total antioxidant capacity (TAC). The quantification of total phenolic content showed that certain types of honey, such as, stingless bee honey and Tualang honey have higher content of phenolic acids and flavonoids, and greater TAC and radical scavenging activity [100,101,102,103] which may indicate more potential in attenuation of oxidative stress in vivo as well. To our knowledge, presently, no study has been conducted comparing antioxidant effects of various types of honey in vivo. Moreover, the composition of the same variant of honey has not been compared from different regions around the world so far, which points toward a likelihood of varied composition of honey obtained from two different countries.

Studies suggest that most of honey’s antioxidant, anti-inflammatory, and neuroprotective properties are due to its phenolic content [104,105]. Phenolic compounds are comprised of four classes of polyphenols: Phenolic acids, flavonoids, stilbenes, and lignans. Out of these, phenolic acids and flavonoids primarily have the potential to act as antioxidants and reduce oxidative stress [29,30,31] and neuroinflammation [32] that are the mediators of insults to the brain in the neurodegenerative diseases [106,107]. However, this review will focus on the effectiveness of flavonoids and phenolic acids on the CNS and in the prevention/treatment of AD pathology. The main phenolic compounds affecting the physiological functioning and/or the pathophysiology of the CNS are stated in Figure 1.

According to the studies on AD models (Refer to Table 1 and Table 2), all mentioned flavonoids and phenolic acids exert antioxidant effects and show neuroprotective activity. All agents, except myricetin, were also found to exhibit anti-inflammatory potential. As myricetin possesses an anti-inflammatory ability against post-ischemic neurodegeneration [108], if tested, it may also display similar potential in AD brain. In addition to the antioxidant and anti-inflammatory potential, most polyphenolic components also proved to attenuate AD pathology by decreasing amyloid deposition, with an exception of kaempferol and chlorogenic acid. Additionally, naringenin, naringin, quercetin, caffeic acid, and ellagic acid also reduce levels of p-tau in AD brain.

## 4. Therapeutic Potential of Flavonoids and Phenolic Acids

Several polyphenols are known to exert protective effects on the nervous system and are suggested to have a role in alleviating symptoms of neurological diseases [159,160,161] including AD [162]. All these phenolic compounds, which are listed in Figure 1, are found to improve cognitive performance in AD pathology, and prevents from cognitive decline when ingested before inception of disease. These compounds are found in different kinds of honey in addition to other sources, and are particularly effective in attenuating oxidative stress along with exerting preventive effects on several other mechanisms in AD pathology. Their potential to reduce AD-induced brain injury is further proven by the microscopic studies and brain assays where they showed neuroprotective effects on the cortex [116,140], hippocampus [124,125], and hypothalamus [119]. Moreover, the consumption of polyphenols resulted in the prevention of hypoperfusion injury [145] with a generalized increment in cell number having normal physiology in the subiculum [122], and hippocampal proper area CA1 [48,136,139,154], CA3, and dentate gyrus [109,120,157], along with preserving normal synapses [112,131,144], the latter is further evident by an increased LTP after polyphenol consumption [154,156,158,163]. Further, the detailed studies of AD-brain animal model showed the potential of the polyphenols to reduce oxidation markers such as MDA [128,153], nitric oxide (NO), and nitrite [112,119,129,130], thereby attenuating free-radical-induced oxidation insults. The observed levels of antioxidants, however, are contrasting, with many studies deducing an elevated level of SOD and catalase, GSH [113,127,140,143,152,155], whereas others concluding decreased expression [115,146,148]. Although the results are contradictory, the studies demonstrating a reduction in the activity of antioxidants claim that this decrease also signifies the attenuation of oxidative injury, which subsequently renders the expression of the anti-inflammatory markers unnecessary. However, despite the discrepancy in the results, all studies conclude that the changes in the expression of these markers lead to reduced oxidative damage and Aβ-plaque accumulation.

Moreover, phenolic compounds can also alter the expression of some critical genes: APP, BACE1, PSEN-1, and Glutathione peroxidase 1 (GPx1). Polyphenols are found to down-regulate the expression of APP [164] and PSEN-1 gene [165], along with either a decrease [142,146,148] or increase in GPx1 expression [116]. As GPx1 is an enzyme that catalyzes the reduction of hydroperoxides and hydrogen peroxide by GSH to attenuate oxidation [166], reduced expression can lead to oxidative injury to cell. Moreover, the expression of BACE1 is also decreased [153] and is thought to be inhibited post-transcription, probably at the protein level [148]. Although not studied yet, a similar down-regulation of PSEN-2 gene expression can be expected by polyphenol consumption [167]. Moreover, polyphenols may also prevent neuritic plaque deposition by increasing α-secretase activity and by reducing cleavage of the APP to amyloidogenic soluble APP-β and β-CTFs [142] and hence, preventing the accumulation of the latter in synapses [148,165]. Taken together, these studies suggest that the polyphenols likely regulate gene expressions to reduce oxidation and formation of Aβ fibrils. Moreover, the polyphenols also increase the expression of the transcription factor Nrf2, which is responsible for regulating the induction of antioxidant genes, thereby improving defense against oxidative injury [139]. To protect the CNS further, the polyphenols reduce the level of pro-inflammatory markers, such as NFκB, TLR4 [139,157], COX-2 [151], MHC class II, TNFα [114,127,146], IL-1α [149], IL-1β [147,151,168], IL-6 [157,169] and increase anti-inflammatory cytokines [115], thereby reducing neuroinflammation. Furthermore, by decreasing neuroinflammation, these substances also attenuate the immunoreactivity of microglia and astroglia in the hippocampus, EC, and amygdala, which is commonly observed in AD neuropathology [115,122,142,146,149,150].

Additionally, the polyphenols reduce tau hyperphosphorylation and subsequent formation of NFT [170] and decrease the deposition of Aβ-plaques [140,156,171]. They also seem to exert a neuroprotective effect by preventing neuronal injury [136,152] and apoptosis [119,136,151] and by regulating the ACh system, where they increase ACh and choline acetyltransferase (ChAT), and decrease acetylcholinesterase (AChE) [119,123] and butyrylcholinesterase (BChE) [153]. These polyphenols’ effects also lead to minimization of deficits of memory and cognitive [172,173,174,175]. As honey contains a number of these polyphenols, its consumption can be expected to have similar potential to prevent and treat CNS pathology in AD. The therapeutic potential of the polyphenols: flavonoids and phenolic acids, are summarized in Table 1 and Table 2, respectively.

The effectiveness of honey in minimizing neurodegeneration is attributed to its neuroprotective effects on the brain [30,34], including the prefrontal cortex [176,177,178] and hippocampus [34,35,179,180]. Honey prevents neurodegeneration by attenuating two main phenomena, which are oxidative stress and neuroinflammation [36,132]. The reduction in neuroinflammation [181] is due to the attenuation of oxidative stress [30,31] and the prevention of free radical-mediated injury to the brain tissue [182,183]. This effect is evident by an increase in the antioxidant enzyme such as SOD and a reduction in the oxidative-stress markers, such as plasma MDA and protein carbonyl in aged brains [176,177]. Subsequently, as the hippocampal pyramidal neurons are highly susceptible to oxidative damage, this reduction of oxidative stress probably rescues them from insult and degeneration [34,182].

Further, along with the hippocampus, injury to the medial prefrontal cortex (mPFC) and piriform cortex is commonly observed in AD, both of which are associated with memory and cognition. Unlike other primary sensory cortices, which are minimally affected by the AD pathology, the piriform cortex is possibly affected even before or along with the development of the cognitive symptoms [184,185,186]. Hence, it is also considered a predictive marker of the conversion to AD [184,187]. Similarly, the damage to the mPFC in AD is evident as defective functioning [188,189] and abnormal connectivity with other associated brain areas [190]. Even though no such study has been undertaken to look at the injuries in these areas, the neuroprotective effects of honey may also rescue the mPFC and piriform cortical injury.

Although researchers widely accept the neuroprotective capacity of honey, we still do not have much data on the effects of honey on the physiology and/or anatomy of the human brain, and, therefore, its potential to act on the CNS is not fully understood to date. It is probably due to the late advent of technologies to study the brain in honey-related research and the limitation to researching human CNS. However, to overcome the limitation of experimental access to the human brains and to understand the possible effect of honey on the microscopic level, the research is now predominantly being carried out in rodents.

## 5. Effects of Honey on Memory, Cognition, and Behavior

Cortical Aβ deposition exerts effects on temporal lobe atrophy and resultant cognitive impairment in individuals with AD [191]. From psychophysiological perspective, cognition, learning, and memory are believed to be mainly determined by the cortico-hippocampal (C-H) circuit’s normal functioning [192,193]. As ACh is the principal neurotransmitter in synapses, the amount of ACh also plays an essential role in learning behavior and cognitive performance [194]. Although relatively constant, the amount of ACh still normally fluctuates, according to the need in the memory processes, such as encoding and retrieval [195]. Essentially, since the integrity of the C-H circuit depends upon the normal physiology of neurons and synapses, the Aβ plaque formation, and therefore AD, may affect the circuit by damaging neurons [196], reducing the number of cholinergic neurons [197], decreasing the ChAT activity [198,199], decreasing ACh release [200], and impairing synapses, which results in defective transmission [72,74]. Surprisingly, ageing and Aβ fibrillogenesis also decrease the AChE activity [201,202]; this finding is unexpected and in contrast to the decreased ACh indicates that the reduction in ACh is likely due to degeneration of the cholinergic neurons along with an increase in another cholinesterase enzyme activity, such as BChE [203] and not due to elevated AChE levels, as the latter is itself hydrolyzed by the former [204,205]. The intake of honey is found to reduce the level of BChE with a further decline in the level of AChE [176,206]. Although the exact mode of action is still not understood, this cholinesterase inhibition, together with neuroprotection, results in improved cognition and memory [207] after honey consumption, as observed in rodents [180,182,208,209] and humans [37,210]. The effects of honey as a nutraceutical agent in improving memory and cognition are further discussed in Table 3 (in rodents) and Table 4 (in humans).

## 6. Honey on Dopaminergic Neurons—Important Players in Memory Deficits in AD

In addition to ACh, dopamine plays a significant role in learning and memory functions. Besides being secreted from dopaminergic neurons (DN) of the Ventral tegmental area (VTA) and substantia nigra pars compacta (SNpC) in the midbrain [216,217], dopamine is also released by locus coeruleus (LC), located in the brainstem, which co-releases dopamine along with noradrenaline [218,219]; this released dopamine from LC innervates CA3 [220], and is thought to be the primary source of supply to the dorsal hippocampus [218,221]; however, a recent study suggests that the midbrain modulate dopaminergic innervation to the dorsal hippocampus and this stimulation is sufficient to arouse aversive memory even in the absence of input from LC [216]. To aid with the understanding of the contrasting source of dopamine, Takeuchi et al. proposed that although the projection of dopaminergic fibers from LC is denser than the midbrain [221], the midbrain and LC both modulate dorsal hippocampus in different kinds of memory consolidation processes [222]. Since the DN in the VTA are the primary site for dopamine synthesis, the DN in the midbrain-hippocampal (M-H) loop are vital in learning, memory formation, and consolidation [223,224]. The DN, and the secreted dopamine, modulate synaptic plasticity and contribute to the LTP in the hippocampus [225], thereby playing an essential part in the genesis and fortification of the episodic [226], aversive [216], and spatial memories [224]. Moreover, dopamine, together with norepinephrine, is crucial for the recognition memory [227,228]. In non-diseased brains, the number of DN and, therefore, the functional connectivity of the midbrain tends to decline with age [229,230], which may appear as deficits in learning and memory [231]. Similarly, as AD is a disease of old age, there is degeneration of DN [232,233]; However, due to the Aβ pathology, probably more damage occurs to the dopaminergic synapses in the M-H loop. Due to the mentioned insult, there is a decrease in dopamine, leading to impaired synaptic plasticity [234,235] and deficits of memory [233,236]. Polyphenols are found to prevent the degeneration of dopaminergic neurons and increase dopamine levels [137,237,238,239,240,241]. Although all these studies discuss the attenuation of neuroinflammation with/without Parkinson’s disease, similar results are expected in the AD model. As the insults to DN and reduced dopamine in AD are recently being studied in detail, more research is encouraged to be conducted on the AD M-H loop to understand the effect of honey and its constituent polyphenols on memory improvement in AD.

## 7. Honey as a Nootropic Agent—Prevention, Treatment, or Both?

In light of the previous research, it is evident that by acting on the CNS and working through various mechanisms, honey acts as a nootropic agent (refer Figure 2 and Figure 3). Now the question arises of the right time to utilize these nootropic properties to alleviate AD symptoms. Another similar issue is understanding whether honey consumption is effective in preventing the development/conversion of mild cognitive impairment into AD, mitigating the damage during ongoing AD disease pathology or reversing the injury done to the brain by AD. Although, to our knowledge, this aspect is not assessed to date, the studies on polyphenols (discussed in Table 1 and Table 2) may suggest some possible effects. The intake of phenolic compounds before initiation of the AD neuropathology is found to halt the progression of the CNS disease, protect neurons, reduce neuroinflammation and oxidative damage, and minimize memory and cognitive deficits, as seen in the studies on the AD-rodent models [48,109,111,112,117]. Likewise, honey ingestion in subjects with developed AD may also cause effects similar to those observed in the polyphenols-treated AD model [121,122,125,126]. Since these polyphenols are abundant in honey, we can expect the same benefits with honey consumption in human subjects.

Although honey is loaded with various kinds of polyphenols [100,102,103], the protective or curative effect of honey can be enhanced further by consuming it in combination with some nutraceutical agent [182,210,242]. Moreover, the synthesis of dimer by combining caffeic acid and ferulic acid [243], and the use of an amino acid (glutamine) conjugated with phenolic acid [244], both of which proved to be more efficient than the polyphenol alone, have paved a path for the likelihood of the advent of similar new combinations with honey that might emerge as the novel therapies for the prevention of AD. Moreover, a mixture of honey with other nutraceutical substances has proven to be effective in AD (for review: [245,246,247]) and can be appraised in prevention and/or management of AD.

In the same notion, the accurate dose of honey to prevent and/or treat AD has not been deduced to date. One of the important reasons of inability to draw conclusions is the fact that most studies are conducted on rodents, with very few studies on human subjects. Moreover, many confounding factors need to be addressed, such as the type of honey to be ingested, the therapeutic dose of honey, the minimum duration of honey intake, stage of AD (if given for treatment). Since few studies mentioned in Table 4 use a formulation of honey and other nootropic agents, another question arises whether combinations with such agents are more effective in terms of dosage and duration in improving human cognition. To our knowledge, currently there are no known studies on primates that observe the benefits of honey on cognition, or sporadic AD which is best modeled by the rhesus monkeys [248]. Comparative studies are encouraged in these areas, with possible usage of primates such as chimpanzees etc., to observe and deduce invaluable conclusions.

## 8. Conclusions and Future Directions

The phenolic compounds prevent damage to the neurons while promoting apoptosis in dysfunctional or cancer cells, which points toward different mechanisms of action in the brain cells than the rest of the body. On the same notion, the polyphenols are believed to exhibit oxidation-promoting properties for review: [249], rendering it necessary to explore the reasons for the switch between pro-oxidant and antioxidant characteristics to utilize these novel qualities appropriately. However, as various polyphenols have anti-amyloid and anti-tau potential, there is a possibility that they function as antioxidants in the cells that contain (or may contain) the Aβ aggregates and NFTs, whereas they promote oxidation in other abnormal cells; this aspect of these substances, as well as that of honey, needs further elaboration.

Some recent studies have demonstrated the potential of flavonoids in prevention of memory decline in elderly individuals [250,251,252,253]. Similarly, the research on animal models of AD (mentioned in Table 1 and Table 2) have shown an effectiveness of polyphenols in prevent and/or treat AD symptoms. However, the source of phenolic compounds in all these studies are variable. Considering the fact that the phenolic compounds can be obtained from various sources, notably fruits, vegetables, beverages, and nuts [254,255], the same polyphenols from distinct sources may have different pharmacological activities, and their polyphenolic potential, especially the capacity to act as an anti-oxidant and anti-inflammatory agent, may also vary accordingly. In the light of these considerations, new studies focusing primarily on the flavonoids and phenolic acids derived from honey are highly encouraged to understand the polyphenolic potential of honey.

Although an increased level of AChE is observed in cerebrospinal fluid (CSF) of AD patients [256], an overall reduction of AChE concentration is found in the AD brain [201,202]. The reduced amount of AChE suggests that AD pathology can be due to the reduction of some neuroprotective variant of AChE (e.g., AChE-R) in the absence of an increase in AChE activity [202]. Taking this into consideration, although the most reliable drug to treat mild to moderate AD is donepezil, an AChE inhibitor, it paradoxically increases the amount of AChE in the CSF [257,258]. These findings suggest that the AChE could have different properties inside the brain and within the CSF, pointing toward the possibility that the effect of honey on memory and cognition is due to neuroprotection with/without some mechanism other than AChE inhibition.

The role of dopaminergic system has been studied for a long time; however, its importance in memory deficits in AD was not clear. With the new studies on the role of dopaminergic system in learning and memory, a decline in dopamine levels with the damage in synapses is observed in AD. Furthermore, it is found that restoration of the dopaminergic levels is associated with improvement of memory deficits in AD [259]. For this purpose, dopamine agonists are being tested and proven to reverse the memory-related symptoms [260,261]. Moreover, the dopamine and its derivatives are found effective to reverse oxidative stress, inflammation, and Aβ load [262]. However, the dopamine replacement methods have given promising results, the effects of polyphenols and honey are still not being elucidated. Considering the polyphenol composition of honey, it can be more effective in treating various aspects of AD neuropathology compared to dopamine alone. On the same note, despite the fact that honey consumption is found to have a miraculous role in treating various diseases [263,264,265], its effectiveness in neurodegenerative diseases is still under evaluation. Having being proven to positively affect cognition and memory, the antioxidant capacity of honey also suggests its potential to manage neurological disorders and neurodegenerative diseases. More studies are needed to be conducted using animal models of neurodegenerative diseases, such as AD, to study the benefits of honey in its treatment and management.

## Figures and Tables

**Figure 1 antioxidants-12-00427-f001:**
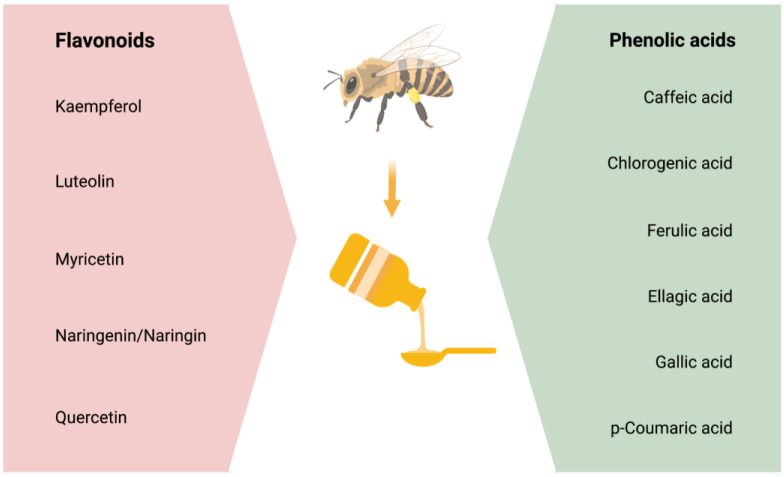
The phenolic compounds, which comprises of flavonoids and phenolic acids.

**Figure 2 antioxidants-12-00427-f002:**
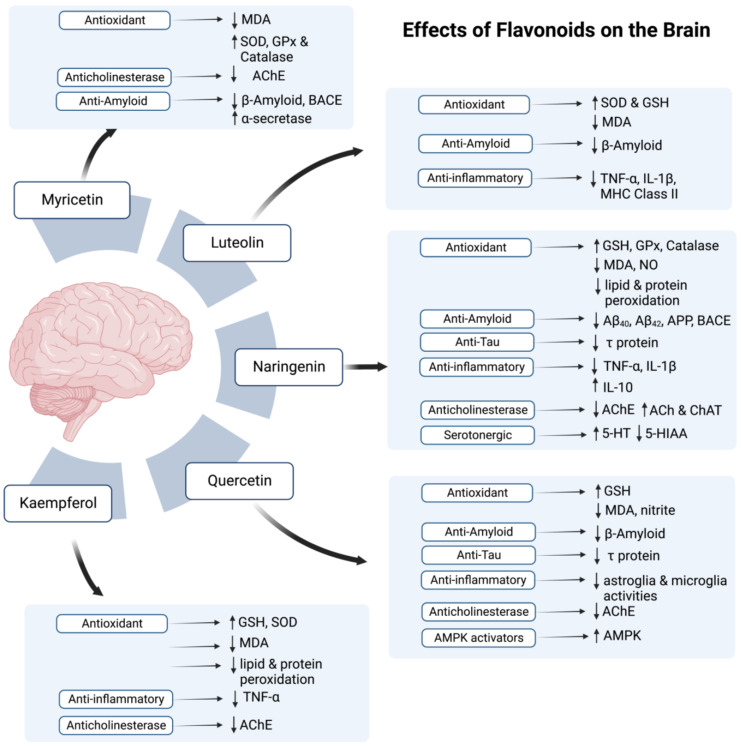
The possible effects of flavonoids in honey on the brain. Symbol (↑) represents increase while (↓) represents decrease.

**Figure 3 antioxidants-12-00427-f003:**
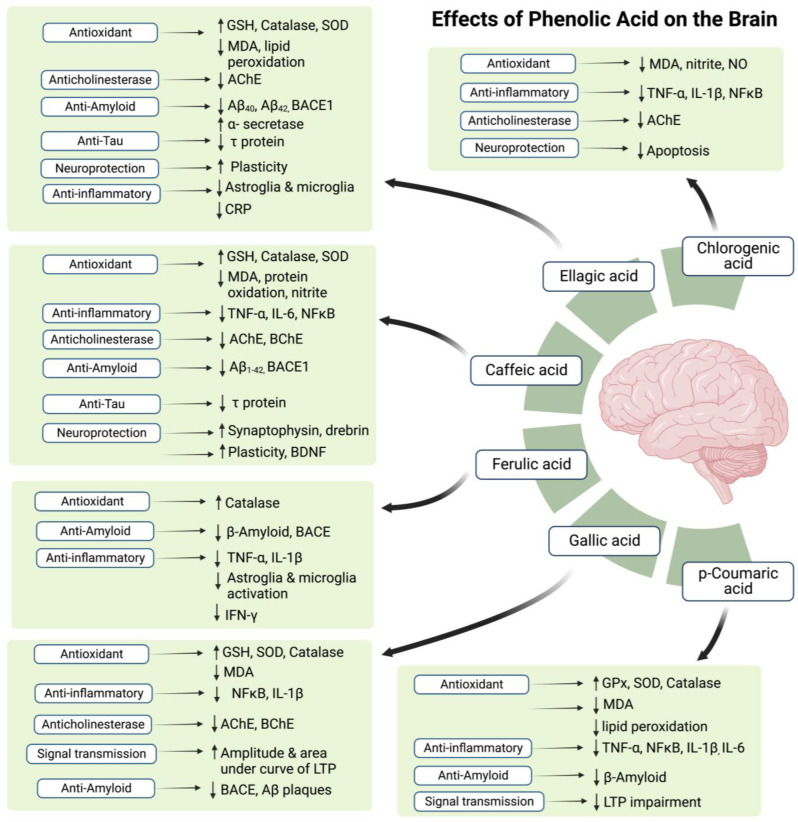
The possible effects of phenolic acids in honey on the brain. Symbol (↑) represents increase while (↓) represents decrease.

**Table 1 antioxidants-12-00427-t001:** Main flavonoid compounds affecting the physiological functioning and/or the pathophysiology of the central nervous system.

Flavonoid Component	Studied Model	Testing Method	Time of Starting Administration	Potential to Act As	Studied Region in Brain	Important Findings of the Study	References
Myricetin	STZ induced AD (Wistar) rat model	Passive avoidance test IHC	1 day before stereotactic surgery (STZ exposure)	Neuroprotective agent	Hippocampus (area CA3)	Myricetin (at 10 mg/kg i.p.,) resulted in a better performance in avoidance test with decreased STL and increased TDC, along with increasing number of intact neurons in CA3 layer.	[109]
	Kunming Mice	MWM test, and brain tissue analysis	Together with i.p. injection of scopolamine	Antioxidant and anti-AChE agent	Hippocampus	Myricetin decreased escape latency and increased time spent in target quadrant, and number of platform crossings. Decreased the amount of MDA while improving antioxidant enzyme activities; it also sustained the concentration of ACh in the hippocampus.	[110]
	Neurons from fetal rat cerebral cortex (E18)	IHC, Immunoblotting, spectroscopy, and activity assays	1 day before Aβ_1–42_ exposure	Anti-amyloid and neuroprotective agent	Not applicable	Myrecetin protects neurons from Aβ_1–42_ induced injury and cell death. It decreases production and aggregation of Aβ_1–42_ and _Aβ1–40_ (only at higher dose) that is also proved by increased activity of α-secretase and decreased activity of BACE1 (in a concentration-dependent manner).	[111]
Luteolin	ICV-STZ induced AD (Wistar) rat model	MWM task and probe tests; IHC	3 days before injection of STZ	Neuroprotective agent	Hippocampus (area CA1)	Luteolin pre-treatment resulted in: Decreased escape latency and travel distance to reach the hidden platform. More time spent in the target quadrant. More pyramidal cells in area CA1.	[112]
	Sprague–Dawley rats (chronic hypoperfusion injury model)	MWM task; Brain tissue analysis	On 5th post-operative day of (bilateral common carotid arter) ligation surgery	Anti-inflammatory, antioxidant, and anti- amyloid agent	Cortex and hippocampus	Luteolin-treated rats showed: Decreased escape latency with more time spent in the target quadrant. Decrease in the MDA level and an elevated SOD activity and the amount of GSH. Decreased levels of TNF-α, IL-1β, and Aβ after luteolin treatment.	[113]
	Adult male Balb/c mice; Murine Neuro.2a, and LPS stimulated BV-2 (murine microglia cell line)	MWM task; Brain tissue analysis	4 weeks before experiment	Anti-inflammatory (aged mice), and neuroprotective (before LPS induction) agent	Hippocampus	Pretreatment with luteolin reduces pro-inflammatory mediators in microglia, therefore, prevents Neuro.2a cell death. Reduction in mRNA levels of IL-1β and MHC class II, and TNFα (only with higher intake). Better performance of aged mice fed with luteolin in MWM task.	[114]
Naringenin/Naringin	High-fat-diet fed SAMP8 mice (a model of AD)	MWM task and Barnes Maze test; Brain tissue analysis	Along with the high-fat diet	Anti-inflammatory, anti-amyloid, anti-tau, and neuroprotective agent	Cortex, hippocampus, and white matter	Naringenin treatment resulted in: Better performance in memory tasks. Suppression of pro-inflammatory markers, and elevation of anti-inflammatory cytokines. Reduction in the levels of soluble and insoluble Aβ_40_, Aβ_42_, APP and BACE1, p-tau, and GSK in hippocampus. Reduced concentration of MDA, NO, and activity of SOD, GSH in cortex (compared to high-fat-diet fed group).	[115]
	AlCl_3_+D-gal induced AD (Wistar) rat model	Behavioral tests; Brain tissue analysis	Two weeks before AlCl_3_+D-gal induction	Antioxidant, anti-AChE, Serotonin- enhancer, and neuroprotective agent	Cortex and hippocampus	Naringenin pre-treatment resulted in: Better performance in memory tasks. Decreased SOD activity and MDA levels, and increased activity of catalase, GPx, and GSH concentration. Increased 5-HT and decreased 5-HIAA concentration. Prevention of neuronal degeneration.	[116]
	Intra- hippocampal Aβ_1–40_ induced (Wistar) rat model	Y-maze, Radial arm maze task, passive avoidance test; Brain tissue analysis	1 h before injecting Aβ_1–40_ bilaterally in the dorsal hippocampus	Antioxidant and neuroprotective agent	Hippocampus	Pre-treatment of rats with naringenin caused: Better performance in behavior tasks. Lower level of MDA, without any significant difference in nitrit4e and SOD concentration. Less DNA fragmentation (considered as a marker of apoptosis) in hippocampi.	[117]
	ICV-STZ induced AD rat model	Passive avoidance test, MWM task; Brain tissue analysis	14 days before ICV-STZ injection	Antioxidant and neuroprotective agent	Hippocampus	Pre-treatment with naringenin resulted in: Better performance in behavioral tests. Attenuation of lipid peroxidation and protein oxidation. Increased level of GSH and increased activity of antioxidant enzymes. Alleviation of Na^+^/K^+^-ATPase activity in hippocampus. Restoration of ChAT neurons while maintaining normal morphology of the neurons in CA1.	[118]
	Hydrocortisone injected AD mice model	MWM task NOR test and step-down test;$$$$$Brain tissue analysis	21 days before hydrocortisone injection	Anti-amyloid, anti-tau, anti-AChE, antioxidant, and neuroprotective agent	Hippocampus and hypothalamus	The results of the pre-treatment of mice with Naringin were: Better performance in behavioral tests. The count, shape and distribution was similar to sham group. Increased expression of estrogen receptor protein. Decreased expression of p-Tau and CDK5 in hippocampus. Inhibition of protein expression of Aβ, APP and BACE1 in hippocampus. Increased ACh and ChAT, and decreased AChE in hippocampus. Decreased levels of MDA and NO, and increased SOD in hippocampus.	[119]
Quercetin	ICR mice subjected to dexamethasone	MWM task	3 h before dexamethasone i.p. injection	Neuroprotective agent	Hippocampus (area CA3 and DG)	More number of cells in DG in quercetin-treated group.	[120]
	ICV-STZ induced AD rat model	MWM task	After 1 week of ICV-STZ induction	Neuroprotective agent	Not mentioned	Decreased escape latency, and increased time spent in target quadrant.	[121]
	Homozygous 3xTg-AD mice	MWM task, elevated plus maze; Brain tissue analysis	Quercetin injected i.p., every 48 h for 3 months in AD mice before experimentation	Anti-amyloid, anti-tau, anti-inflammatory and neuroprotective agent	Subiculum, area CA1, entorrhinal cortex and amygdala	Quercertin-treated group showed: An increased cell density in subiculum. Decreased Aβ and tau fibrillary tangles deposition and in CA1, subiculum, and amygdala. Significantly reduced astroglial and microglial immunoreactivity in the CA1 hippocampal area, the entorrhinal cortex, and the amygdala. Improved memory in behavioral tests.	[122]
	I.C.-STZ induced (Swiss) albino mice	MWM task Passive avoidance test; Brain tissue analysis	Just after I.C.- STZ injection	Antioxidant and anti-AChE agent	Whole brain (homogenate)	Reduced mean latency in MWM task and increased TLT. Reduction in MDA and nitrite levels, and inhibition of AChE activity (with higher dose of quercetin). Increased GSH levels in quercetin-treated mice.	[123]
	ICR mice subjected to TMT-induced neuronal deficits	Y-maze and passive avoidance test; Brain tissue analysis	21 days before the TMT induction	Antioxidant and anti-AChE agent	Whole brain (homogenate)	Quercetin pre-treatment resulted in: Improved performance in behavioral tests. Inhibitory effect on AChE, with inhibition of lipid peroxidation (at a higher dose of Quercetin). Antioxidant and radical scavenging ability shown by ABTS and FRAP assays.	[124]
	APPswe/PS1dE9 (C57/BL) transgenic mice	NOR test, MWM test; Brain tissue analysis	16 weeks before sacrifice	Antioxidant, anti-amyloid, and neuroprotective agent	Hippocampus and cortex	Mice treated with quercetin showed an increased recognition index in NOR test, decreased escape latency in MWM task. Quercetin increases AMPK, prevents the formation of amyloid plaques, and alleviates hippocampal-mitochondria dysfunction.	[125]
Kaempferol	Transgenic Aβ flies (DS model)	Climbing assay; Brain tissue analysis	30 days before behavioral tests	Antioxidant, anti-AChE, and neuroprotective agent	Whole brain (homogenate)	Dose-dependent increase in GSH content, and decrease in LPO, PC, GST, and AChE activity after kaempferol treatment compared with unexposed Aβ-flies. Decreased apoptosis (evident by lower level of caspase enzymes) compared with the unexposed Aβ-flies.	[126]
	Ovariectomized ICV-STZ induced AD (Wistar) rat model	MWM test; Brain tissue analysis	On the same day as 2nd dose of STZ, and continued for 21 days	Antioxidant and anti-inflammatory agent	Hippocampus	Kaempferol consumption caused: Reversal of STZ-induced cognitive dysfunction. Enhanced hippocampal SOD and GSH levels. Reduced levels of inflammatory markers MDA and TNF-α.	[127]
	ICV-STZ induced AD (Wistar) rat model						[128]

MWM = Morris Water Maze; NOR = novel object recognition; TLT = transfer latency time (i.e., the time taken to move from the open arm into any covered arm with four legs); ICR = Institute of Cancer Research; STZ = streptozotocin; ICV = intracerebroventricular; I.C. = intracerebral; IHC = immunohistochemistry; STL = step-through latency; TDL = time spent in the dark chamber; i.p. = intraperitoneal; TMT = trimethyltin; MDA = malondialdehyde; ABTS = 2,2′-azino-bis(3-ethylbenzothiazoline-6-sulfonic acid); 5-HIAA = 5-hydroxyindoleacetic acid; FRAP = ferric reducing antioxidant power; GSH = reduced glutatione; GPx = glutathione peroxidase; GST = glutathione-S-transferase; LPO = lipid peroxidation; PC = protein carbonyl content; ROS = reactive oxygen species; CDK5 = cyclin-dependent kinase 5; SAMP8 = senescence-accelerated mouse prone-8; NO = nitric oxide; ChAT = choline acetyltransferase; GSK3β = glycogen synthase kinase-3β; LPS = lipopolysaccharide; AMPK = AMP-activated protein kinase.

**Table 2 antioxidants-12-00427-t002:** Main phenolic acid compounds affecting the physiological functioning and/or the pathophysiology of the central nervous system.

Phenolic Acid Component	Studied Model	Testing Method	Time of Starting Administration	Potential to Act As	Studied Region in Brain	Important Findings of Study	References
Caffeic acid	ICV-STZ induced AD (Wistar) rat model	MWM, NOR test and spontaneous locomotor activity; Brain tissue analysis	1 h after first dose of ICV-STZ	Antioxidant and anti-AChE agent	Cerebral cortex and hippocampus	Caffeic acid-treated rats showed: Dose-dependent improvement in STZ-induced cognitive dysfunction. Better memory (with a higher dose of caffeic acid), and more time spent in the target quadrant. Dose-dependent attenuation of MDA, PC, and nitrite levels. Restoration of depleted GSH and inhibition of AChE activity.	[129]
	AlCl_3_-induced AD (Wistar) rat model	MWM; Brain tissue analysis	20 days after AlCl_3_ (daily) injection	Antioxidant and anti-AChE agent	Whole brain (homogenate)	Reversal of AlCl_3_ –induced memory deficits. Inhibition of AChE activity, and nitrite levels in brain. Increased levels of catalase, GSH, and GST in caffeic acid-treated group.	[130]
	High-fat-diet-induced AD (Sprague-Dawley) rat model	MWM; Brain tissue analysis	Along with the high-fat diet	Antioxidant, anti-amyloid and$$$$$anti-tau agent	Cerebral cortex and hippocampus	Reversal of memory deficits in caffeic acid group. Increased SOD, and decreased level of APP expression, β-Amyloid_(1–42)_ content and BACE1 levels. Decreased in p-Tau (Thr181) expression. Increased synaptophysin expression in cortex, and drebrin expression after caffeic acid treatment.	[131]
	High carbohydrate high fructose (HCHF) diet induced metabolic syndrome (Wistar) rat model	Brain tissue analysis	After consumption of HCFC diet for 8 weeks	Anti-inflammatory agent	Hippocampus (area CA1 and DG)	Reduced TNF-α levels, and higher BDNF concentration compared with HCHF-only fed group.	[132]
	Intrahippocampally- Aβ_1–40_-induced AD (Sprague-Dawley) rat model	MWM; Brain tissue analysis	After injecting Aβ_1–40_	Antioxidant, anti-inflammatory, anti-AChE, and neuroprotective agent	Hippocampus	Caffeic acid-treated group showed: Decreased escape latency and mean path length, and more time spent in target quadrant. Increased synaptophysin expression, and decreased AChE activity. Decreased nitrite along with increased catalase and GSH levels. Reduced NFκB-p65 expression with decreased activity of IL-6 and TNF-α. Decreased p53 and P-p38 MAPK expression.	[133]
	Wistar rats (whole brain in-vitro)	Not applicable	Added to the supernatant of the homogenate	Antioxidant, anti-AChE, and anti-BChE agent	Whole brain homogenate	Addition of caffeic acid caused: Dose-dependant inhibition of AChE and BChE. Dose-dependant decrease in MDA content. High total antioxidant capacity and radical scavenging ability.	[134]
	i.p. D-gal induced aging (Sprague-Dawley) rat model	Novel Object Location, NOR; Brain tissue analysis	Along with D-gal	Neuroprotective agent	Hippocampus	Co-treatment with caffeic acid displayed: Dose-dependent attenuation of memory impairment. Enhanced hippocampal neurogenesis by attenuation of reduced cell proliferation and increased survival of mature neurons.	[135]
Chlorogenic acid	APP/PS1 double transgenic mice	MWM;Brain tissue analysis	At 3-month of age	Neuroprotective agent	Brain (including histological evaluation of hippocampal CA1 area)	Chlorogenic acid-treated mice showed: Decrease in escape latency with more time spent in the target quadrant. Cholorogenic acid protected against Aβ_25–35_ induced autophagy and promoted lysosomal function in APP/PS1 brain while restoring normal morphology of neurons in area CA1.	[136]
	C57BL/6 mice + Primary neuro-glia cultures	Brain tissue analysis + Neuro-glia analysis and assays	7 days before LPS injection + 2 h before incubation with LPS	Anti-inflammatory and neuroprotective agent	Substantia nigra	Pre-treatment with chlorogenic acid: Attenuated LPS-induced IL-1β and TNFα expression in Substantia nigra. Inhibited nitrite and nitric oxide production along with attenuation of TNFα expression and NFκB signaling in LPS-stimulated microglia. Protected dopaminergic neurons from microglia-mediated LPS toxicity.	[137]
	Scopolamine-induced AD (ICR) mice model	Y-maze test, passive avoidance test, MWM; Brain tissue analysis	30-min before scopolamine injection	Antioxidant, and anti-AChE agent	Whole brain (homogenate), and frontal cortex and hippocampus (homogenate)	Cholorogenic acid pre-treatment resulted in: Prevention of scopolamine-induced (short-term and long-term) learning and memory deficits. Inhibition of AChE activity and MDA levels in hippocampus (at all tested doses) and frontal cortex(only at higher dose).	[138]
Ellagic acid	Intrahippocampal microinjection Aβ_25–35_ induced AD (Wistar) rat model	NOR, Y-maze, passive avoidance and radial arm maze tasks; Brain tissue analysis	One week before Aβ-induction surgery	Anti-inflammatory, antioxidant, anti-AChE, and neuroprotective agent	Hippocampus (including histological evaluation of CA1 area)	Ellagic acid pre-treatment caused: Improved discrimination ratio and memory performance. Decreased MDA with an increase in GSH (at both doses), and catalase (only at higher dose). Restored NFκB and nuclear/cytoplasmic ratio for Nrf2 (at both doses), and decreased TLR4 expression(only at higher dose). Decreased level of AChE activity along with prevention of decline of CA1 neuronal count (at both doses).	[139]
	AlCl_3_-induced AD (Wistar albino) rat model	NOR test; Brain tissue analysis	After stopping AlCl_3_	Antioxidant, anti-amyloid, anti-tau, and neuroprotective agent	Whole brain	The results of ellagic acid treatment were: Better memory in NOR test compared with untreated AD group. Decreased lipid peroxidation along with increased levels of catalase, GSH, and total antioxidant capacity. Improved neuronal morphology and lowering of amyloid and tau burden in cerebral cortex. Co-treatment of ellagic acid with ellagic acid-loaded nanoparticles was more effective in mitigating all the behavioral and brain abnormalities.	[140]
	APP/PS1 double-transgenic mice	MWM; Brain tissue analysis	1 week after acclimatization	Anti-amyloid, anti-tau, and neuroprotective agent	Hippocampus	Improved learning and memory in the ellagic acid-treated group. More number of neurons with reduced expression level of caspase-3 in hippocampus. Decreased Aβ plaque deposition along with reduced levels of both Aβ_40_ and Aβ_42_ which is also confirmed by reduction in pThr668-APP and BACE1 expression. Down-regulation of p-tau (pSer199-tau and pSer396-tau) by mediating AKT/GSK3β signaling pathway (increasing pSer473-AKT and lowering pTyr216-GSK3β).	[141]
		NOR, Y-maze, radial arm water- maze tasks; Brain tissue analysis	At 12 months of age	Antioxidant, anti-inflammatory, and anti-amyloid agent	Whole brain (including study on EC, RSC, and hippocampus)	Treatment with ellagic acid resulted in: Complete reversal of learning and memory impairments and (anxiety-like) behavioral abnormalities. Upregulation of α-secretase and downregulation of BACE1 with reduced Aβ-plaque deposition (both Aβ_1–40_ and Aβ_1–42_), and CAA. Decreased number of immunoreactive glia (astroglia and microglia) with reduced expression of SOD1 and GPx1.	[142]
	Oral AlCl_3_- induced AD (Wistar) rat model	NOR test; Brain tissue analysis	4 weeks after the beginning of oral AlCl_3_ dosage	Antioxidant, anti-amyloid, anti-tau, and neuroprotective agent	EC	Ellagic acid-treated group showed: Improved discrimination index in NOR test. Increased serum SOD (due to upregulated gene expression), GSH levels and higher mean total antioxidant capacity with decreased levels of TBRS (products of lipid peroxidation). Restored thickness of EC with more neurons having normal morphology. Down-regulation of APP and caspase-3 expression with reduced load of NFTs.	[143]
	ICV-STZ induced AD (Wistar) rat model	Radial arm maze and Y-maze tasks; Brain tissue analysis	1 day after STZ administration	Antioxidant, anti-inflammatory, anti-amyloid, and neuroprotective agent	Cerebral cortex (homogenate), EC and hippocampus proper (area CA1, CA2, CA3, and DG)	Ellagic acid treatment caused: Improved memory and cognitive scores. Reduced levels of MDA and CRP together with elevated GSH and catalase activity. Neurons having normal morphology with decreased Aβ-plaque burden in EC and hippocampus proper. Reduction of immunoreactive astroglia and elevation of synaptophysin levels.	[144]
Ferulic acid	APP/PS1 (transgenic) mice	MWM task; Brain tissue analysis	In AD mice of 6 months age	Anti-amyloid, neurovascular protective agent	Whole brain (including study on cerebral cortex and hippocampus)	Ferulic acid treatment effects on APP/PS1 mice were: Restoration of learning and memory impairment. Increased density of whole-brain blood vessels (including hippocampus) with prevention of reduction of diameter of hippocampal capillaries and, therefore, cerebral blood flow. Reduction of Aβ plaque deposition in hippocampus(both Aβ_1–42_ and Aβ_1–40_) and cortex along with attenuated BACE1 activity. Reduced microglia aggregates surrounding Aβ plaques.	[145]
		NOR, Y-maze, radial arm-water maze tasks; Brain tissue analysis	In 1-year-old AD- mice model	Anti-inflammatory, antioxidant, and anti-amyloid agent	Whole brain (including study on RSC, EC, hippocampus)	Treatment with Ferulic acid resulted in: Reduction in cerebral amyloidosis and CAA. Decreased reactive gliosis with reduced expression of SOD1, GPx1, TNF-α and IL-1β. Elevated synaptophysin immunoreactivity in area CA1 and EC. Moreover, the combination therapy of Ferulic acid with epigallocatechin-3-gallate was more effective and completely reversed all the behavioral and brain abnormalities.	[146]
		NOR and Y-maze tasks; Brain tissue analysis	In 6-month old AD mice	Anti-amyloid, anti-inflammatory agent	Frontal cortex and hippocampus	Ferulic acid treatment: Improved memory performance in NOR after low-dose (5.3 mg/kg/day) treatment, whereas treatment at a higher dose (16 mg/kg/day) was ineffective. Reduction in cortical Aβ_1–40_ and Aβ_1–42_ levels (more effective at lower dose) with alleviation of IL-1β levels (at both doses).	[147]
	PSAPP mice (AD-model)	NOR, Y-maze and MWM tasks; Brain tissue analysis	In 6-month-old AD mice	Antioxidant, anti-inflammatory and anti-amyloid agent	Cingulate cortex, EC and hippocampus + Whole brain (homogenate)	Ferulic acid-treated PSAPP mice displayed: Remediation of learning, memory and behavior impairment. Reduction of cerebral amyloid burden and CAA by attenuating both Aβ_1–40_ and Aβ_1–42_ levels along with decreased BACE1 activity (at the translational/protein level). Attenuation of glial activation with reducing expression of TNF-α, IL-1β, SOD1, catalase and GPx1.	[148]
	ICR mice (ICV-induced Aβ_1–42_ AD-model)	Brain tissue analysis	4 weeks before ICV injection of Aβ_1–42_	Antioxidant and anti-inflammatory agent	Hippocampus	Ferulic acid pre-treatment mitigated oxidative stress and neuroinflammation by blocking astroglial activation evident by double staining of 3-nitrotyrosine and endothelial nitric oxide synthase immunoreactive cells with GFAP.	[149]
		Brain tissue analysis	4 weeks before ICV injection of Aβ_1–42_	Antioxidant and anti-inflammatory agent	Hippocampus	Ferulic acid pre-treatment inhibited microglial activation evident by blocking of OX-42 (marker of activated microglia) and IFN-γ immunoreactivity.	[150]
Gallic acid	ICV Aβ_1–42_ induced AD (ICR) mice model	Y-maze and passive avoidance test; Brain tissue analysis	3-weeks before ICV injection of Aβ	Anti-inflammatory and neuroprotective agent	Whole brain (homogenate), Cerebral cortex and hippocampus	Gallic acid pre-treatment: Prevented Aβ-induced cognitive deficits. Restored cytokine (iNOS and COX-2) levels in the cerebral cortex and hippocampus induced with Aβ and decreased neuronal apoptosis. Inhibited nuclear translocation and acetylation of NFκB and prevented subsequent IL-1β release in mice brain.	[151]
	Oral AlCl_3_-induced AD (Wistar) rat model	Y-maze and MWM tests; Brain tissue analysis	Together with AlCl_3_	Antioxidant and neuroprotective agent	Hippocampus	Gallic acid co-ingestion group showed: Improved learning and memory indices in behavioral tests. Elevated levels of catalase, SOD and GSH with more neurons in hippocampus.	[152]
	APP: BACE [high] transgenic Drosophila AD-model	Brain homogenate analysis	5 days before sacrifice	Antioxidant, anti-inflammatory, anti-AChE, and anti-BChE, and anti-amyloid agent	Whole brain (homogenate)	Gallic acid caused: Decreased BACE1 and Cholinesterase (AChE and BChE) activity. Reduced MDA levels and increased catalase activity together with amelioration of reactive oxygen species burden. Raised total thiol content.	[153]
	Intrahippocampal Aβ_1–42_ induced AD (Wistar) rat model	Electrophysiological analysis; Brain tissue analysis	The 2nd day after intrahippocampal injection	Anti-amyloid	Hippocampus (area CA1 and DG)	Improved amplitude and area under curve of LTP as recorded from DG in gallic acid-treated flies. Reduced burden of Aβ plaques in area CA1 in treated group.	[154]
	ICV-STZ induced AD (Wistar) rat model	Passive avoidance and MWM tests; Brain tissue analysis	5 days before ICV-STZ injection	Antioxidant agent	Cerebral cortex and hippocampus	Pre-treatment with Gallic acid resulted in: Improved learning and memory in behavioral tests. Decreased levels of MDA and increased total thiol levels with restoration of SOD, GPx and catalase activity.	[155]
p-Coumaric acid (p-CA)	AlCl_3_-induced AD (Wistar) rat model	Passive avoidance test; Electrophysiological analysis; Histological analysis	1-h prior to AlCl_3_-induction	Anti-amyloid and neuroprotective agent	Hippocampus	p-CA pretreatment resulted in:Improved memory retrieval in behavioral test. Mitigation of LTP impairment that is evident by increased amplitude and area under curve for the population spike, and the field excitatory postsynaptic potentials slope in electrophysiological recordings. Reduced burden of Aβ-plaques in DG.	[156]
		OFT, elevated plus maze, MWM, and forced swimming tests; Brain tissue analysis		Antioxidant, anti-inflammatory, and neuroprotective agent	Cerebral cortex and hippocampus (histological studies on area CA1, CA3 and DG)	p-CA pre-treatment improved memory, decreased anxiety and depression-like behavior, and increased exploratory activity. p-CA increased SOD, GPx, and Catalase activities and decreased MDA NFκB, TNF-α, IL-1β, and IL-6 levels. Increased number of intact neurons.	[157]
	Scopolamine-induced AD (Sprague-Dawley) rat model	Passive avoidance and MWM test Electrophysiological analysis	1-h before scopolamine administration	Neuroprotective agent	Hippocampus (area CA1)	p-CA causes the following changes: Increases amplitude of LTP with increased field excitatory postsynaptic potentials in electrophysiological studies in p-CA group. Inhibition of NMDA receptor and AMPA receptor blockade and resultant increase in LTP. Inhibits muscarinic receptor blockade which is proposed as the likely reason behind memory improvement in behavioral test.	[158]

EC = entorhinal cortex; RSC = retrosplenal cortex; DG = dentate gyrus; CC = cerebral cortex; CA = cornu ammonis; NFTs = neurofibrillary tangles; CAA = cerebral amyloid angiopathy; LTP = long-term potentiation; NOR = novel object recognition; OFT = open field test; LPS = lipopolysaccharide; TLR4 = Toll-like receptor 4, Nrf2 = nuclear factor (erythroid-derived) 2; NFκB = nuclear factor-kappa B; SOD = superoxide dismutase; PC = protein carbonyl content; BACE = β-Site APP-cleaving enzyme; GPx = glutathione peroxidase; TBRS = thiobarbituric-acid-reactive substances; AlCl_3_ = aluminum chloride; IFN-γ-gamma interferon; NMDA receptor = N-methyl-D-aspartate receptor; AMPA receptor = α-amino-3-hydroxy-5-methyl-4-isoxazolepropionic acid receptor.5. Effect of honey as a neuroprotective agent.

**Table 3 antioxidants-12-00427-t003:** Effects of honey intake on memory and cognition in animal model.

Studied Model	Dosage	Duration of Exposure	Findings	Reference
Wistar rats induced with metabolic syndrome effects (MetS) by feeding high carbohydrate high fructose (HCHF) diet	15 mL of Kelulut honey dissolved in 15 mL of distilled water given at 0.1 mL/kg of body weight daily	35 days (after 16 weeks of HCFC diet)	Reduced anxiety compared with the MetS group Enhanced memory efficiency than both control and MetS groups Increased number of pyramidal cells in the hippocampus compared with the MetS group	[34]
Female Swiss albino mice (age = 2.5 months)	A total concentration of 750 mg/kg and 2000 mg/kg among groups—with 0.3 mL of Stingless bee honey dilution forced-fed daily	7 days (acute) and 35 days (semichronic)	Improvement in learning and spatial reference memory	[211]
Stressed ovariectomized Sprague–Dawley rats (approximately 8 weeks old)	0.2 g/kg body weight daily of Tualang honey diluted with 1ml of distilled water	18 days (started 3 days before stress induction)	Decreased anxiety-like behavior in the stressed ovariectomized rats (StOE) that is comparable to β-estradiol (E2) treatment	[29]
			Improved short-term and long-term memory in the StOE rats Increased pyramidal cells in CA2, CA3, and DG of hippocampus in the StOE rats Results were comparable to that after E2 treatment	[180]
Sprague–Dawley rats (approximately 2 months old)	Experimental diet of containing 100 g/kg honeydew honey that was available ad libitum	3, 6, 9, and 12 months	Improved spatial memory and decreased anxiety-like behavior	[208]
Sprague–Dawley rats (2 months old)	0.2 g/kg body weight Tualang honey dissolved in distilled water/daily	35 days	Improved short-term and long-term memory Decreased depressive-like symptoms	[209]
Sprague–Dawley rats (16 months old)	200 mg/kg body weight of Tualang honey/daily	28 days (started 14 days prior to stress procedures)	Improvement in both short-term and long-term memory More Nissl-positive cells in the mPFC and hippocampus Greater number of pyramidal cells in the mPFC and hippocampus exhibiting normal shape and structure	[176]
Male Sprague–Dawley rats Young (2 months old), and aged (16 months old)	200 mg/kg body weight of Tualang honey/daily	28 days (started 14 days prior to stress procedures)		[177]
Wistar rats	0.5, 1.0, and 2.0 g/kg body weight	Singly dose (1 h before behavioral tests)	Honey, in a dose-dependent manner, ameliorates the anxiety-like behavior and possibly also acts as an anti-depressant	[212]
Swiss albino mice Young (3–4 months), aged (12–15 months)	The formulation containing honey (400 mL), ghee (800 mL) and gold (288 mg) was given at the dose of 30 mg/kg/daily	15 days	The formulation intake improved learning and memory in young and aged mice Decreased activity of AChE in brain	[213]

**Table 4 antioxidants-12-00427-t004:** Effects of honey consumption on memory and cognition in humans.

Subjects	Type of Study	Dosage	Duration of Exposure	Findings	Reference
Mild cognitively impaired, and Cognitively intact controls (all over 65 years old)	Randomized, placebo control, double-blind	1 tablespoon daily	5 years—tested every 6 months	About 28% of the placebo-given subjects, while less than 7% of the honey-ingested subjects developed dementia.	[37]
Postmenopausal women (aged between 45 and 60 years)	Cohort	20 g Tualang honey (sachet) daily	16 weeks	Improved learning and memory scores in the auditory verbal learning test	[38]
Patients diagnosed with mood disorders and candidates for electroconvulsive therapy (ECT) (aged > 18 years)	Randomized, double-blind	9 g of herbal combination of *Crocus sativus*, *Cyperus rotundus*, and honey/twice daily	40 days (after initiation of ECT)	Improvement of ECT-induced memory improvement, especially after one to two months of the last ECT session.	[214]
Depressed elderly individuals (aged 60 or more)	Crossover randomized	25 g Talbinah honey in 100 mL of water daily	6 weeks (3 weeks + 1 week break + 3 weeks)	Improvement in depression, stress, and mood disturbances scores.	[215]
Patients diagnosed with mild to moderate major neurocognitive disorder	Randomized, double-blind	10 g Asparagalus honey with 1000 mg of sedge and 60 mg of saffron extracts daily	3 months	Improved attention, memory and cognition compared with the placebo-given group.	[210]

## Data Availability

No new data were created or analyzed in this study. Data sharing is not applicable to this article.

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
