# Peer review of "Honey and Alzheimer’s Disease—Current Understanding and Future Prospects"

_antioxidants, 2023, doi:10.3390/antiox12020427_

Round 1
Reviewer 1 Report
The work examines the neuroprotective, antioxidant and anti-inflammatory role of honey in a multivariable pathology such as AD by analyzing different aspects and many bibliographic works.
Many data are present in the literature both for dementia and for different oncological, cardiovascular, neuro-psychological pathologies ... Other nutrients contain chemical agents similar to those analyzed for honey (coffee, hops, cocoa...) which reflect their "properties beneficial" but, as the same authors conclude, much is still to be investigated to understand the real mechanisms of action, benefits and methods of administration.
As far as animal models are concerned, it is also difficult to obtain animal models of sporadic forms of AD in order to be able to study the more common forms of the disease.
As far as animal models are concerned, it is difficult to obtain specific animal models for the sporadic forms of AD, which are the most common forms of the disease; so, how do the authors think it is possible to study the potential benefits of these substances in "ad hoc" animal models?
Authors should also correct some punctuation errors and grammatical slips (Ex. caption fig2 , paragraph of the conclusions,..); furthermore some abbreviations are mentioned in the text and then subsequently described in the figures (EC, GPx1,...)
Author Response
As far as animal models are concerned, it is difficult to obtain specific animal models for the sporadic forms of AD, which are the most common forms of the disease; so, how do the authors think it is possible to study the potential benefits of these substances in "ad hoc" animal models?
We thank the reviewer for reviewing our manuscript. Here are the responses to your queries:
Animal models provide us with the ability to do preclinical testing in vivo, assess the general toxicity of novel medicines, and to conduct cognitive tests. We acknowledge and fully agree with the reviewer that many of the current animal model used in research did not truly reflects the majority of AD cases which is the sporadic form of AD. Since it is impossible to study preclinical testing in human, the next best candidates have been identified in primates, specifically rhesus monkey where they found that the Aβ levels accumulate with age. However, tauopathy was found to be rare in these species. Therefore, it is critical to recognise that none of the available models replicate all features of human AD and thus cannot be considered representative models of AD as a whole disease. However, the use of currently available animal models can provide a means to answer vital questions about AD pathophysiology, as long as one has a thorough understanding of the chosen model and its inherent limitations to ensure that experimental results can be translated to human AD (Drummond and Wisniewski, 2017). It is not the intention of our manuscript to explain on this matter, as the aim of our paper is to describe the achievement of honey research in AD.
Drummond E, Wisniewski T. Alzheimer's disease: experimental models and reality. Acta Neuropathol. 2017 Feb;133(2):155-175. doi: 10.1007/s00401-016-1662-x.
Reviewer 2 Report
In this review, the authors underlying the positive effects of honey intake on memory and cognition in various animal models.
Particularly, the review is focalized on Alzheimer disease, as the current treatment aims to treat the symptoms and to slow the disease progression, as there is no real cure. Continuous researches for new nutraceutical agent or medicine to help prevent and cure AD pathology are always going on. In this quest, honey has emerged as a powerful nootropic agent. Numerous studies have demonstrated that the high flavonoids and phenolic acids content in honey exerts its antioxidant, anti-inflammatory and neuroprotective properties.
In general, the work is well articulated, where the tables are a fundamental part of this research in the literature.
Questions:
· Do all types of honey intake improve memory and cognition? or do we have to see the honey typologies? Or, there are Different Polyphenols in Various Types of Honey. Which is the most suitable?
· Where does it have to come from the type of honey (i.e. acacia, sunflowers..)? and which country?
· In the paragraph 2 (2. Pathophysiology and clinical picture of Alzheimer’s disease) the authors reported: “…..Likewise, the neuroinflammation is evident by an elevation of the pro-inflammatory markers, such as IL-1α, IL-1β, IL-6 [82,83], TNFα and NFκB [84,85] with an accompanied reduction in some anti-inflammatory cytokines[86,87].” Please explain which cytokines are reduced? And why?
· In the paragraph 3: Each polyphenol and flavonoid indicated in Fig 1 has its own peculiar characteristic: antioxidant, and/or anti-inflammatory capacity. Insert in detail the descriptions.
· In the paragraph 4: there’re potential side effects of honey? And/or with drug interaction?
· In the paragraph 8: can you hypothesize a dose for prevention, and/or for treatment? Are there anthropology/evolution studies where they were monitored from a cognitive/behavioral point of view in animals that eat honey? (e.g. chimpanzee?)
· Insert a diagram/figure that shows the effects of the various components of honey in Alzheimer's pathology (animal model and human)
Minor point:
Line 38: insert the references
and Check always the acronyms in all text.
Best regards
Author Response
We thank the reviewer for reviewing our manuscript. Here are the responses to your queries:
Q1: Do all types of honey intake improve memory and cognition? or do we have to see the honey typologies? Or, there are Different Polyphenols in Various Types of Honey. Which is the most suitable?
Where does it have to come from the type of honey (i.e. acacia, sunflowers..)? and which country?
Ans: Due to the mentioned factors, various types of honey are different in composition of polyphenolic compounds (Cheung Y. et al., 2019; Sousa J.M. et al., 2016), and therefore, polyphenolic activity and total antioxidant capacity (TAC). The quantification of total phenolic content showed that certain types of honey, such as, Stingless bee honey and Tualang honey have higher content of phenolic acids and flavonoids, and greater TAC and radical scavenging activity (Ranneh Y. et al., 2018; Kek S. P. et al., 2014; Monirruzaman M. et al., 2013; Kishore R.K. et al., 2011) which may indicate more potential in attenuation of oxidative stress in vivo as well. To our knowledge, presently, no study has been conducted comparing antioxidant effects of various types of honey in vivo. Moreover, the composition of the same variant of honey has not been compared from different regions around the world so far, which points towards a likelihood of varied composition of honey obtained from two different countries.
We have incorporated this answer in the manuscript under heading No 3. Honey and its powerful ingredients – The phenolic compounds, under paragraph 1.
Q2: In the paragraph 2 (2. Pathophysiology and clinical picture of Alzheimer’s disease) the authors reported: “…..Likewise, the neuroinflammation is evident by an elevation of the pro-inflammatory markers, such as IL-1α, IL-1β, IL-6 [82,83], TNFα and NFκB [84,85] with an accompanied reduction in some anti-inflammatory cytokines[86,87].” Please explain which cytokines are reduced? And why?
Ans: Likewise, perpetual neuroinflammation marked by an imbalance in the inflammatory cytokines is also evident by the over-expression of the pro-inflammatory markers, such as IL-1α, IL-1β, IL-6 [82,83], TNFα and NFκB [84,85],and an accompanied under-expression of some anti-inflammatory cytokines, such as IL-4 and IL-10 [86,87]. Besides, the reduced level of anti-inflammatory cytokines further leads to the uninhibited activity of pro-inflammatory cytokines(Su, P. et al., 2016; Wang, P. et al., 1995) and results in more neuronal damage (Su, F. et al., 2016).
We have also incorporated this answer in the manuscript under heading No 2. Pathophysiology and clinical picture of Alzheimer’s disease, under paragraph 4.
Q3: In the paragraph 3: Each polyphenol and flavonoid indicated in Fig 1 has its own peculiar characteristic: antioxidant, and/or anti-inflammatory capacity. Insert in detail the descriptions.
Ans: According to the studies on AD models (Refer to Tables 1 and 2), all mentioned flavonoids and phenolic acids exert antioxidant effects and show neuroprotective activity. All agents, except Myricetin, were also found to exhibit anti-inflammatory potential. As Myrecetin possesses an anti-inflammatory ability against post-ischemic neurodegeneration (Pluta R. et al., 2021), it may also display similar potential in the AD brain upon being tested. In addition to the antioxidant and anti-inflammatory potential, most polyphenolic components also proved to attenuate AD pathology by decreasing amyloid deposition, except for Kaempferol and Chlorogenic acid. Additionally, Naringenin, Naringin, Quercetin, Caffeic acid and Ellagic acid also reduce levels of p-tau in the AD brain.
We have incorporated this answer in the manuscript under heading No 3. Honey and its powerful ingredients – The phenolic compounds, under paragraph 3 (new paragraph).
Q4: In the paragraph 4: there’re potential side effects of honey? And/or with drug interaction?
Ans: The only known adverse effect of honey is hypersensitivity, which ranges from a mild hypersensitivity reaction to severe anaphylactic shock. All cases reported presented a history of intake of honey or honey-containing food. The body's allergic reaction was later proven by increased serum-specific immunoglobulin E (IgE) antibodies in patients' sera. The affected patient usually presents to the emergency department with a swollen lip, urticaria, and angioedema with/without breathing difficulty after honey ingestion. No drug interactions have been reported for such honey-related hypersensitivity reactions.
We have incorporated this answer in the manuscript under heading No 3. Honey and its powerful ingredients – The phenolic compounds, under paragraph 3
Q5: In the paragraph 8: can you hypothesize a dose for prevention, and/or for treatment? Are there anthropology/evolution studies where they were monitored from a cognitive/behavioral point of view in animals that eat honey? (e.g. chimpanzee?)
Ans: In the same notion, the accurate dose of honey to prevent and/or treat AD has not been deduced to date. One of the important reasons of inability to draw conclusions is the fact that most studies are conducted on rodents, with very few studies on human subjects. Moreover, many confounding factors need to be addressed, such as the type of honey to be ingested, the therapeutic dose of honey, the minimum duration of honey intake, stage of AD (if given for treatment). Since few studies mentioned in Table 4 use a formulation of honey and other nootropic agents, another question arises whether combinations with such agents are more effective in terms of dosage and duration in improving human cognition. To our knowledge, currently there are no known studies on primates that observe the benefits of honey on cognition, or sporadic AD which is best modeled by the rhesus monkeys (Drummond and Wisniewski, 2017). Comparative studies are encouraged in these areas, with possible usage of primates like chimpanzees etc., to observe and deduce invaluable conclusions.
Minor point:
Line 38: insert the references
The statement here is referring to the next sentences, and the references has already been added there.
and Check always the acronyms in all text.
Best regards
Round 2
Reviewer 1 Report
The authors speculated to clarify my request of the previous review
Thank you